# A hybrid Voronoi Tessellation/Genetic Algorithm Approach for the Deployment of Drone-Based Nodes of a Self-Organizing Wireless Sensor Network (WSN) in Unknown And GPS Denied Environments

**Khouloud Eledlebi** [1], **Hanno Hildmann** [2], **Dymitr Ruta** [3] and **A. F. Isakovic** [4,5,*]

1    Department of Electrical Engineering, Khalifa University, 127788 Abu Dhabi, UAE; khouloud.edlebi@ku.ac.ae
2    Intelligent Autonomous Systems Group, Netherlands Organization for Applied Scientific Research (TNO), Oude Waalsdorperweg 63, 2597 AK The Hague, The Netherlands; hanno.hildmann@tno.nl
3    Data Science Group, Emirates ICT Innovation Centre (EBTIC), 127788 Abu Dhabi, UAE; dymitr.ruta@ku.ac.ae
4    Physics and Astronomy Department, Colgate University, Oak Dr. 13, Hamilton, NY 13346, USA
5    CNF, Cornell University, Ithaca, NY 14853, USA
*    Correspondence: aisakovic@colgate.edu

**Abstract:** Using autonomously operating mobile sensor nodes to form adaptive wireless sensor networks has great potential for monitoring applications in the real world. Especially in, e.g., disaster response scenarios—that is, when the environment is potentially unsafe and unknown—drones can offer fast access and provide crucial intelligence to rescue forces due the fact that they—unlike humans—are expendable and can operate in 3D space, often allowing them to ignore rubble and blocked passages. Among the practical issues faced are the optimizing of device–device communication, the deployment process and the limited power supply for the devices and the hardware they carry. To address these challenges a host of literature is available, proposing, e.g., the use of nature-inspired approaches. In this field, our own work (bio-inspired self-organizing network, BISON, which uses Voronoi tessellations) achieved promising results. In our previous approach the wireless sensors network (WSN) nodes were using knowledge about their coverage areas center of gravity, something which a drone would not automatically know. To address this, we augment BISON with a genetic algorithm (GA), which has the benefit of further improving network deployment time and overall coverage. Our evaluations show, unsurprisingly, an increase in energy cost. Two variations of our proposed GA-BISON deployment strategies are presented and compared, along with the impact of the GA. Counter-intuitively, performance and robustness increase in the presence of noise.

**Keywords:** Voronoi centroids; genetic algorithm; particle swarm optimization; drones; drone swarms; swarming; swarm intelligence; wireless sensor networks; self-organization; self-optimization; energy aware; noise coherence; position-navigation-timing; GPS denied

## 1. Introduction

Drones have become an affordable consumer item. They are increasingly considered as candidates for mobile/aerial urban sensing platforms [1]. Projected advances in cloud computing, wireless sensors and networked unmanned systems motivate this claim further, and indeed, in the literature UAVs are increasingly being suggested as flexible mobile platforms [2] used for applications as wide as general monitoring [3–6], monitoring and surveillance [7–9], to provide situational awareness support [10], detect victims or their life-signs [4] and participate in search and rescue (SAR) or aerial tracking [7].

This is especially applicable in the direct aftermath of a natural of man-made disaster as these events often result in partial or total loss of communication and data-collection infrastructure.

UAV swarms [1,11]—that is, a number of individual devices cooperating to form a collective operational unit—have been used as mobile sensor networks [2] for monitoring and tracking civilians, personnel and victims [4] and for surveillance tasks in general [7]. A recent literature study considered a number of surveys [12–17] identifying application areas for UAVs and UAV swarms [18]. Of all considered wireless access networks, many emphasized the disaster response domain (cf. Table 1).

**Table 1.** Recommended surveys in the literature focusing on one or more application areas for UAVs and UAV swarms, as identified by [18] and discussed in [19]. **WAN** = *wireless access networks*, **RS** = *remote sensing*, **RTM** = *real-time monitoring*, **SAR** = *search and rescue*, **GL** = *goods delivery/logistics*, **INT** = *surveillance* and **SI** = *structural inspection*. WAN is considered by all as a relevant area.

| Reference | WAN | RS | RTM | SAR | GL | INT | SI |
|---|---|---|---|---|---|---|---|
| [12] | ✓ | | | ✓ | ✓ | | ✓ |
| [13] | ✓ | ✓ | | | | | |
| [14] | ✓ | | ✓ | | ✓ | ✓ | |
| [15] | ✓ | | | | | | |
| [16] | ✓ | | | | | | |
| [17] | ✓ | | | | | | |
| [18] | ✓ | ✓ | ✓ | ✓ | ✓ | ✓ | ✓ |

Drones are increasingly considered as platforms for mobile sensing and actuation [19,20]. With cellular network coverage [21–30] and aerial sensing [2–4,7,17,29,31–43] among the most popular applications. As a general rule, disaster response operations have to be implemented within the first 72 h [10,34,44] so as to avoid a dramatic increase in loss of life/economic damage. Any technology that can speed up the e.g., assessment of structural damage of buildings or the localization of victims through indoor exploration [45] can reduce the cost of life and bodily harm.

In this field, UAVs have a lot of potential [10,19]: drone based wireless sensors networks (WSNs) are developed for a wide variety of applications (e.g., health-care and environmental monitoring [5,46–51]). With advances in the field of autonomous intelligent systems, and given the progress made with mobile cyber-physical systems, building and deploying *mobile* WSNs (where individual nodes can move and are endowed with computational power to enable information driven self-organization and on-board reasoning and decision making) is becoming increasingly common [17,21–43].

Due to this, a variety of challenges (such as limitations in energy, computing power and memory, and difficulties in the deployment process) are emerging. In the design of approaches and strategies to handle a variety self-organizing network constraints for deployment and reallocation procedures, focus is placed on energy-consumption management and ensuring network connectivity.

Considering nature-inspired techniques to enable a WSN to operate in the real-world seems logical, and indeed this is a popular approach in the literature. Combining a number of such approaches to complement each other to improve network survival chances are of great interest to the community.

In the literature, Voronoi tessellations are preferred over other geometrical structures to reduce the computational complexity [52,53]. In this paper, we augment BISON (bio-inspired self-organizing network), our previously developed, Voronoi-based algorithm [54,55] with a genetic algorithm and evaluate the resulting effects on network discovery rate and performance.

### 1.1. Motivation, Objectives and Goals

Merging Voronoi-based approaches with other nature-inspired algorithms such as *particle swarm optimization* (PSO) [52,56–62], *genetic algorithms* (GAs) [63–77] and *ant colony optimization* (ACO) [78–81] is not uncommon (cf. Table 3). There are many examples where this has been shown to potentially

provide stability in the network with coverage optimization, distributed sensor nodes more uniformly, and ultimately, enhance the lifetime of the network [58,59].

The literature indicates that both, PSO and GAs improve energy consumption and WSN deployment time comparably [60]. However, as GAs are more compatible with the locally available information that sensor nodes rely on during their deployment process (cf. Section 2.2.2), we focus our work on a GA. A full scale comparison between hybrid BISON+GA and BISON+PSO is beyond the scope of this paper but could be considered for future work. As initial explorations indicated that GA is more promising, we studied different variations for augmenting BISON with a GA under varying parameters such as population size, number of iterations and mutation rate.

- The work presented in this paper is guided by three main goals:

  1. To design a novel hybrid collaboration between our previous BISON algorithm and a GA, with the aim of further enhancing the performance of nodes in a WSN;
  2. To evaluate and compare the WSN service and deployment performances of different GA/BISON combinations;
  3. To study the robustness and resilience of GA-BISON approaches when deployed in dynamic environments and to compare the outcomes of this investigation with the performance of the original BISON algorithm.

- The main contributions of this article over the state of the art are:

  1. We introduce a hybrid approach that augments the Voronoi based BISON algorithm with a genetic algorithm (Algorithm 1 on page 8). Two variations of the new approach are proposed; the respective algorithms (Algorithms 2 and 3 are provided on pages 10 and 11.
  2. We measure and compare the effects of adding the GA to BISON in general and compared the original to the two (new) hybrid algorithms. Our findings, indicating when and to which end each algorithm should best be used, are summarized in Table 5 on page 20.
  3. We evaluate all three approaches for simple and complex scenarios (cf. Section 4.2 for performance measures; cf. Section 4.3 for parameter space exploration). The results are presented and discussed in Section 5.1.
  4. As the presence of noise has unexpected beneficial effects, we also performed a noise coherence analysis, the outcome of which is provided in Section 5.2.

We close this section by providing the reader with an overview over the symbols used (Table 2).

**Table 2.** Symbols and notation used in this article.

| Symbol | Meaning/Usage | Introduced/Used in |
|---|---|---|
| $\mathcal{S}$ | the set of nodes in the WSN | Algorithms 2 and 3 on pages 10, 11 |
| $N_{n_i}$ | the numbers of neighboring nodes for node $i$ | Equation (1) on page 10 |
| $r$ | the number of nodes to consider | Algorithm 1 on page 8 |
| $n_i$ | the position (location) of a node $i$ | Section 2.3 (or Figure 1 on page 7) |
| $d(n, n')$ | the distance between two locations $n$ and $n'$ | Equation (1) on page 10 |
| $R_{C_i}$ | the communication range of node $i$ | Section 4.1 |
| $R_{S_i}$ | the sensing range of node $i$ | Section 2.3 (or Figure 1 on page 7) |
| $A_{S_i}$ | the area covered by $R_{S_i}$ | Section 2.3 (or Figure 1 on page 7) |
| $A_{V_i}$ | the Voronoi area of node $i$ | Section 2.3 (or Figure 1 on page 7) |
| *num_sol* | the number of solutions to generate per node | Algorithm 1 on page 8 |
| $\alpha$ | control parameter: number of iterations | Algorithm 1 on page 8 |
| $\beta$ | control parameter: mutation rate | Algorithm 1 on page 8 |
| *pop_size* | control parameter: population size | Algorithm 1 on page 8 |
| *minCoverage* | threshold (coverage) to trigger the use of the GA | Algorithms 2 and 3 on pages 10, 11 |
| *stopCDT* | termination criterion based on node movement | Algorithms 2 and 3 on pages 10, 11 |
| *stopCount* | termination criterion based in coverage improvement | Algorithms 2 and 3 on pages 10, 11 |
| PAC | Area coverage (in %) | Section 4.2 |
| CDT | The distance travelled by all nodes in the WSN | Section 4.2 |
| $cnm(t)$ | the sum of all node movements during a time step | Section 4.2 (to calculate CDT, cf. Equation (3)) |

## 2. Background and Preliminaries

### 2.1. Problem Statement and Outlook for the Approach

Many challenges arise when aiming to cover an indoor environment optimally with a fully connected, drone-swarm-based WSN. This is especially the case when the deployment time is an important factor. When the geometry of the target environment is known, then this problem does not appear to be very difficult, with the main difficulty to be resolved being how to physically move ever smaller and versatile mobile nodes to the geometrically derivable known optimal locations at the least energy expense, and how to do so as quickly as possible while remaining collision-free [56,82]. However, the situation complicates enormously when the target environment is not fully known. Yet, given the real-world application of disaster response and recovery, this is a fundamental property of the application. In such cases the optimal locations or the number of required nodes are not known in advance. Due to this, we can expect to deploy the network incrementally with mechanisms in place to expand the deployment and to optimize the locations of the deployed drones and to do so on-the-fly as the swarm discovers the environment through local sensing. Such self-expansion poses in turn a challenge to the efficiency of deployment: even if the network is eventually able to determine the optimal number of participating nodes and their optimal locations, achieving this in effective (ideally minimal) deployment time is challenging. Finally, once all these individual challenges are resolved to an acceptable degree, the challenge to define, for each node in the WSN, the optimal communication strategies remains to be addressed.

The evidence in existing research and many natural world examples suggests that the critical requirements for a successful deployment of the WSN should be: (a) to follow a simple, fast, cost-efficient and decentralized self-organizing process, (b) to autonomously discover—and adapt to—the topology of the target environment and (c) to avoid obstacles discovered, and to do so in real-time. A well-designed WSN should allow sensors to operate autonomously based on local sensing and through the operation policy reduced to interactions, only with the local neighbors who can sense each other. An automated adaptation requirement forces the nodes' policy to be completely independent of their initial positions but be only the function of the locally-perceived structure of the space and its possible obstacles. Previous work on Voronoi-based algorithms considered a random distribution of sensor nodes, with most algorithms relying on prior knowledge about the environment and node locations, while assuming a guaranteed connectivity among the sensor nodes. In contrast, our approach sets to evolve the locations of homogeneous sensor nodes in unfamiliar and obstructed noisy environments. The proposed deployment process will follow iterative steps at which the nodes continuously re-examine their relative positions among peers, walls and other obstacles in order to ensure each *mWatt* of energy invested in changing their location is towards the direction of the emptiest uncovered space, while maintaining connectivity in the network.

Few existing algorithms deal with handling obstacles in the coverage environment, but those that do predominantly implement it through repulsive forces: nodes may be repelled by other nodes or by obstacles in the environment. In many cases the environment is first scanned to determine the obstacle location and shape before distributing the nodes. This is, however, very costly (energy expense), and inefficient, since unknown environments might take a very long time to be discovered and mapped exhaustively [83,84]. An approach that can interact with the environment even if it is only partially observed can be expected to deploy faster (though without guarantee for optimal deployment).

An approach capable of using local and partial knowledge only will furthermore be able to operate in dynamic environments. In BISON, nodes only scan their local environment and identify nearby objects / nodes. Their obstacle-trimmed Voronoi regions are calculated using this subjective information, as is their resulting movement (aiming to position them closer to their updated Voronoi centroid). In the literature, this local information about the position and movement of neighbouring nodes and obstacles is often assumed to be perfect. This is a strong simplification, as measurements in the real world will always be subject to error and noise. Some methods propose using anchor nodes of known locations to predict the locations of all other nodes [78,79], or by adding a (random) uncertainty value to the measured distances between the nodes and their neighbors [65].

In the presented work the impact of noisy localization is simulated by injecting realistic Gaussian-distributed random noise to each sensed location and demonstrated rather strong robustness of our deployment process to quite substantial levels of noise. In fact, on the course of a noise coherence analysis, we demonstrate how underlying nodes' kinematics are affected by environmental noise that incidentally leads to improvements in the network coverage in certain scenarios. We also demonstrate that our WSN deployment proposition code-named as BISON is sustainable with a built in mechanism for efficient removal of dead/tired nodes and replacement of them with the new ones. A hybrid implementation of BISON and a genetic algorithm is also considered for further improvement in the coverage and energy consumption perspectives. Performance is evaluated in terms of several metrics that indicate very fast convergence at the fairly small deployment cost. All the investigated components, comparatively analyzed in terms of various performance metrics and previously unseen properties, validate our model as a very promising WSN deployment process.

### 2.2. Nature-Inspired Heuristics to Improve Voronoi Tessellations

As shown in Table 3, nature-inspired approaches have been shown to provide adaptive multi-objective mechanisms with fewer complications and demands [85–88]. As the field of nature-inspired heuristics is large and growing, an overview is outside the scope of this paper. Therefore, we will only briefly mention some representatives of such approaches (which we have used in the past) and refer the interested reader or developer to [89], made available online for free by the author (http://www.cleveralgorithms.com/).

### 2.2.1. Frequently Used Nature-Inspired Heuristics for WSN

Popular heuristics for WSN deployment and optimization are: particle swarm optimization (PSO) [56,57,63] and genetic algorithms (GAs) [64,82] (both of which provide enhancements in nodes positions to prolong the network lifetime through reducing energy consumption and the number of steps required for WSN deployment), Voronoi-based approaches [78,83,84] (which can maximize coverage by detecting coverage holes in the WSN) and ant colony optimization (ACO) [65,79] (which can increase a network's lifetime by optimizing routing paths between nodes, thereby reducing their energy consumption). Table 3, below, provides a brief comparison between these approaches.

**Table 3.** A quick introduction to—and an overview of—the available literature on combining Voronoi tessellations with three popular nature-inspired heuristics: *ant colony optimization* (**ACO**), *genetic algorithms* (**GA**) or *particle swarm optimization* (**PSO**). We distinguish between *how the two approaches are applied* (the **method**), *what each does* (the **process**) and *the effect(s) they have* (the **impact**).

| | Voronoi + ACO [78–81] | Voronoi + GA [63–77] | Voronoi + PSO [52,56–62] |
|---|---|---|---|
| **METHOD** | Voronoi tessellations are used to determine *all possible* paths in the network. Assuming the cost (length) of all paths is known, ACO is then used to identify *the shortest* path. | Voronoi tessellations are used to detect coverage holes while the GA/PSO is used to reduce energy consumption and to maximize the lifetime of the network. Specifically, ... ... GA achieves this by generating new candidate solutions (locations), while ... | ... PSO does generate virtual points, i.e., Voronoi vertices or random boundary points. |
| **PROCESS** | This works by assigning weight values to the Voronoi edges to guide the search. | This effectively changes the node distribution as well as adds extra mobile nodes. | Node location and velocity are changed and a node's sensing range can be changed. |
| **IMPACT** | This impacts node distribution, weight values as well as the evaluation function. | The objective function is influenced, as are coverage holes (and GA parameters). | Affects virtual points, node location/velocity and the best local/global solution. |

Particle Swarm Optimization (PSO)

Merging PSO with Voronoi tessellations is considered among the major processes of maximizing the coverage and prolonging the lifetime of WSN [52,53,59,61,66]. The authors in [53,61] used PSO to determine the next position of the sensor nodes, while Voronoi tessellations are generated to detect the coverage holes and evaluate the fitness function. These results either rely on a centralized node to gather information about all nodes' locations, do the calculations and suggest new locations/new sensing ranges for each sensor node; or consider each node to have prior knowledge about its own location and thaose of all other nodes. Depending on the application requirements, the objective function may focus on either minimizing the energy consumption or minimizing the gaps in coverage.

When minimizing the energy consumption, PSO diagnoses the sensor node with the highest energy consumption and suggests changing the sensing range (as opposed to changing the node's location). When minimizing coverage gaps, both the sensing range and the location are considered.

Genetic Algorithms (GA)

A significant amount of literature suggests combining a GA with Voronoi tessellations to address the WSN deployment and re-allocation problem. This has successfully been used to enhance WSN coverage and network lifetime [58,67,68,80] but these approaches all rely heavily on the GA using a developed objective function to evaluate the candidate solutions and to select the next positions of nodes with the Voronoi tessellations used only to detect coverage holes. The latter is achieved using prior knowledge about the nodes' locations; GA-generated candidate solutions are initialized inside the respective Voronoi regions. WSN nodes are moved away from or toward their neighbors to reduce these holes [67,68]; new nodes are added to locations specified by the GA to cover the holes [58].

Ant Colony Optimization (ACO)

ACO has been used together with Voronoi-based approaches, mostly to optimize the path planning for communication between nodes within the network. ACO continuously explores connection paths in a WSN to find and adapt to the shortest path [69,81].

2.2.2. Motivating Our Decision to Chose GAs over PSO

Both GAs and PSO are used in WSN to reduce the energy consumption in the network and enhance the lifetime, through managing the routing paths and finding the best decisions to be taken by nodes. We consider it to be beneficial to implement a GA with our BISON algorithm. This is because evolutionary algorithms, specifically GAs, have the flexibility of generating new candidate solutions inside the confined Voronoi region and to do so on the basis of information available locally to each node [88]. In contrast, PSO often requires distributing the nodes' best local and global solution to all nodes, which means it relies on guaranteed connectivity and heavy communication between all the nodes. Furthermore, a central node might be needed to collect all information, perform calculations and then distribute the results. With this in mind, we motivate using a GA as follows:

1.　GAs generate candidate solutions based exclusively on information available to a node locally.
2.　A GA only requires communication with its neighboring WSN nodes (as opposed to all).

*2.3. BISON Algorithm: A Quick Review and Overview*

We previously [54,55] introduced BISON (bio-inspired self-organizing network), an approach using Voronoi tessellations to iteratively deploy homogeneous mobile nodes into unknown environments to establish a WSN. Our approach does not rely on special injection conditions and can maximize coverage while maintaining or re-establishing WSN connectivity automatically. Nodes draw on locally available information only and can autonomously determine the direction to move to new positions. The process is described in the next paragraph and illustrated in Figure 1, below. Note that in this process, nodes only require information available locally to determine new positions to move to.

Any node *i* can calculate the perpendicular bisectors for the lines connecting it to its neighbors. As shown in Figure 1b, these lines form a polygon. In addition, $n_i$, the drone's location, is at the center of a circular region (shown in Figure 1a) with the node's sensing range $R_{S_i}$ as the radius.

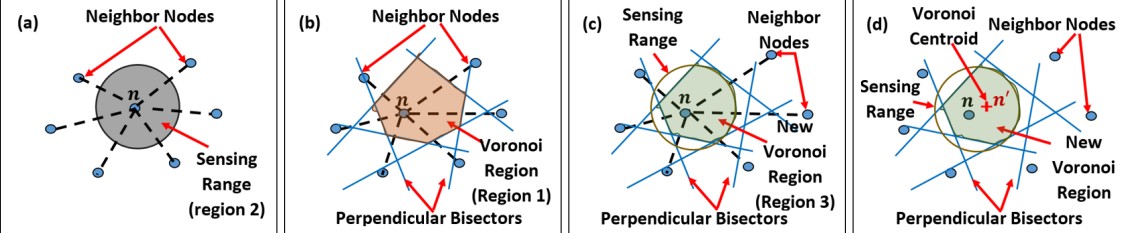

**Figure 1.** Simplification of the working principle behind the BISON algorithm. In a wireless sensors network (WSN), any drone acting as a node *i* in the WSN can calculate two areas: the area $A_{S_i}$ defined by its the node's sensing range ($R_S$), shown in panel (**a**), and its Voronoi polygon $A_{V_i}$ resulting from overlapping sensing ranges with neighboring nodes, shown in panel (**b**). Combining these two results in the actual area covered by the node; see panel (**c**). For any such polygon we can calculate $n'$; i.e., the hypothetical position of the drone if it were at its center of gravity; see panel (**d**). In BISON, the difference between *n* and $n'$ is used to continuously drive them to move toward their centers of gravity [70], and through this, explore any accessible space and achieve coverage and connectivity [54,55]. The basic algorithm, enhanced with genetic algorithms (cf. Figure 2), Section 3.2; the BISON *fixed nodes*, Section 3.3; and the BISON *conditional* approach, Section 3.4), differ mainly in how this is achieved.

The union of the areas shown in (a) and (b) constitutes the area covered by the drones; if this area is not bounded by perpendicular bisectors (as it is the case in Figure 1c) the node is not at its area's center of gravity, as shown as $n'$ in Figure 1d. Moving towards $n'$ allows the node to explore its environment [70]. Note that, while nodes without close neighbors remain stationary in this process, their eventual re-connection to the WSN is ensured through the injection of additional nodes into the environment. This is triggered whenever any of the already deployed nodes fails to report to the injection point, which indicates that either the injection point itself is not covered by the WSN or that a node has become disconnected by moving too far away). In a bounded (finite) environment, this process will deterministically lead to full (as far as possible) coverage and network connectivity.

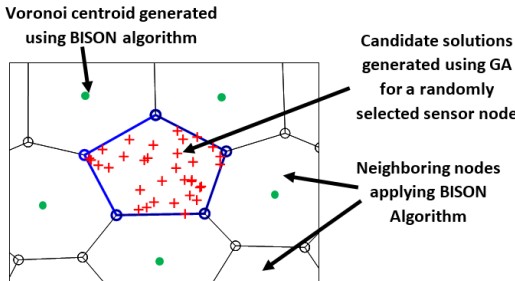

**Figure 2.** Illustrating the difference between movement destination in BISON and the GA: the red crosses are randomly selected candidate locations that are evaluated by the node before one is chosen as the new target location. For comparison, the green dots are the deterministically calculated centroids the neighboring cells moved toward if they used BISON. Figure 3 shows this in the context of a swarm of nodes which is partly of filling an inside area: nodes at the outer edge may not have a neighboring (touching) node at all, but as the number of nodes increases the circular coverage becomes a polygon.

In a computer simulation, obstacle detection is trivial (if we want it to be). When engineering a real world implementation, obstacles can be sensed based on the reflection of radiated signals [90] from the sensor nodes toward their surrounding environment or having bumper sensors [91].

Since we are dealing with a (simplified) circular sensing range, omnidirectional sensor nodes are assumed to transmit signals along 360°. Any signal blocking/distorting obstacle within the sensing range will reflect the signal to some extent, enabling the device to infer the distance to the object and its (approximate) location. New Voronoi regions are calculated from detected obstacle boundaries and the sensing range and the neighboring bisectors.

## 3. The GA-BISON Approach

BISON relies on the drives of nodes to move towards *"their" center of gravity* (cf. Figure 1d). In Biology, the laws of physics provide this force but drones are not inherently aware of the equivalent of this for their Voronoi tesselations. Instead, this has to be calculated during run-time and on-board. All values required to do so can be measured/estimated using existing technologies.

---

**Algorithm 1: basic Genetic Algorithm.** Pseudo-code for the general genetic algorithm, used by BISON *FixedNodes* (in line 9 of Algorithm 2) and *conditional* (in line 10 of Algorithm 3). The algorithm takes the following input: $r$ (the number of nodes to consider), $num\_sol$ (the number of solutions to generate per node), $pop\_size$ (the population size), $\alpha$ (the number of iterations) and $\beta$ (the mutation rate). The output of the algorithm is the solution $s$.

---

1　**for** *each of the (selected) $r$ nodes $n_i$* **do**
2　　generate $v_{n_i}$, the Voronoi region for $n_i$;
3　　$\mathcal{S}olutions \leftarrow$ populate with $num\_sol$ random solutions (i.e., locations inside $v_{n_i}$);
4　　$\mathcal{C}andidates \leftarrow$ select $pop\_size$ locations from $\mathcal{S}olutions$;
5　　$extra = num\_sol - pop\_size$;
6　　**for** $\alpha$ *iterations* **do**
7　　　**for** *node* $\in \mathcal{C}andidates$ **do**
8　　　　find Voronoi region $V_{node}$ for *node*;
9　　　　calculate area $A_{V_{node}}$;
10　　　　calculate area of sensing range $A_{S_{node}}$;
11　　　　using Equation (1), determine the value of $\mathrm{obj}_{fun}(node)$;
12　　　**end**
13　　　$pop_1 \leftarrow$ select best $\frac{pop\_size}{2}$ candidates *node* $\in \mathcal{C}andidates$ (using $\mathrm{obj}_{fun}(node)$);
14　　　**for** $j = 1 : 2 : size\ pop_1$ **do**
15　　　　$\mathrm{offs}(j) \leftarrow (\mathrm{X}pop_1(j), \mathrm{Y}pop_1(j+1))$;
16　　　　$\mathrm{offs}(j+1) \leftarrow (\mathrm{X}pop_1(j+1), \mathrm{Y}pop_1(j))$;
17　　　　save $\mathrm{offs}(j)$ and $\mathrm{offs}(j+1)$ to $pop_2$;
18　　　**end**
19　　　**if** *size extra* $< pop\_size$ **then**
20　　　　$mut_{point} \leftarrow \beta \times pop\_size$ number of random points from *extra*;
21　　　**end**
22　　　**else**
23　　　　$mut_{point} \leftarrow \beta \times pop\_size$ number of random points from *extra* with size $pop\_size$;
24　　　**end**
25　　　$mut_{point_{ind}} \leftarrow$ index of $mut_{point}$;
26　　　mutate $pop\_size(mut_{point_{ind}})$ with $mut_{point}$;
27　　　$\mathcal{C}andidates \leftarrow$;
28　　**end**
29　　evaluate $\mathrm{obj}_{fun}$ for $pop\_size$;
30　　$s \leftarrow max_{pop\_size}(\mathrm{obj}_{fun})$;
31　　return $s$;
32　**end**

---

As this requires calculations to be performed anyway (incurring computational cost) it stands to reason to investigate whether these calculations can be simplified or whether using something other than the center of gravity can yield comparable results. As it turns out, using a GA to determine good candidate locations for the drone to move to (cf. Figure 2) does not only match the performance of BISON, it shows the potential to discover even better locations for the nodes to move to next, resulting in better the convergence of the network as a whole [71–74,80].

### 3.1. Basic Modeling Choices

### 3.1.1. Basic Modeling Decisions for Drone-Swarm Based Indoor WSN Deployment

With regard to using a drone swarm to explore an indoor environment, the entry point into this environment can play a crucial role. The decision on this matter is very likely going to be dependant on the specific environment encountered in each scenario. For our investigations we assume that there is such an entry point given, that there is only one such point and that—for the sake of this investigation—it is found at the coordinates $(0,0)$ (cf. Figure 3). We believe that this choice is fair one to make for our use case scenario, because drone deployment will start from the entrance into the environment and we can allocate the coordinates $(0,0)$ without loss of generality. As for the position in the environment, we argue that being in a corner or along a wall makes little difference. Placement inside the environment (representing drone injection through the floor or the roof) might make a difference but this is not considered in our performance evaluation and left for future work.

The rate of injecting new drones is determined by the proximity of the drone added most recently to the injection point: once this distance exceeds the sensing range ($R_S$) of the drone, a new drone is powered up. The same happens when there is at least one drone that has no current connection to the WSN (indicating that it has drifted outside the sensing range of all its peers). Basically, by requiring (a) the drones to form a connected network and (b) the injection point to be covered by the WSN (and due to the fact that drones will *push outwards* and into other drones, in turn causing them to move away from the injection point) we have an indicator on whether the number of drones in the swarm suffices to provide complete coverage. Whenever this is detected we add one more drone to the swarm. See Figure 3 for an example of drones pushing into an environment to form a connected network.

### 3.1.2. Basic Modeling Choices for the Generic Algorithm

With regard to the use of genetic algorithms to determine a new location for a drone to move to, two fundamental decisions have to be made: (1) *where to look* for new candidate locations (since we are not simply using the center of gravity) and (2) *how to compare* potential candidate solutions. In this section, we first address these two choices (Section 3.1) and then propose a generic GA-enhanced BISON algorithm (Algorithm 1, Section 3.2) as well as two refined versions thereof where the GA is either applied to a initially fixed number of nodes (Algorithm 2, Section 3.3) or where this is done conditional on the state of affairs at the moment the algorithm is run (Algorithm 3, Section 3.4).

1. WHERE to look for (new) candidate locations: The GA will only consider locations inside the node's confined Voronoi region. The motivation for this is that, as shown in Figure 1, we are certain that these locations are known to the node. Given this, we can simply generate random points within the area as potential candidate solutions (depicted by red crosses in Figure 2).

2. HOW to compare the quality of candidate locations: While most objective functions in the literature focus on a single optimization problem, such as coverage, connectivity, or energy consumption [73–75] we use multi-objective optimization. The few multi-optimization objective functions in the literature, e.g., combine coverage with either energy consumption (distance traveled) or with inter-node connectivity [76,77,92]. Our objective function combines three parameters: *coverage* (measured—for each candidate location—as the ratio between the area of the Voronoi region $A_{V_i}$ and the area defined by the sensing range $A_{S_i}$ of node $n_i$ [93]), *connectivity* (the number of neighboring nodes), and *energy cost* (measured as the distance to travel).

---

**Algorithm 2: GA-BISON (*fixed nodes*).** The algorithm takes the following input: a set of nodes $\mathcal{S}$ to which to apply the algorithm to, *minCoverage* (a threshold for minimum coverage to be achieved before the algorithm is used, initially set to 50), *stopCDT* (a termination criterion on the observed node movement, set to $\frac{R_S}{100}$) and *stopCount* (a second termination criterion on iterative coverage improvements, set to 15).

---

1  **while** *(stability > termination|| count < stopCount)* **do**
2     **if** *(exist disconnected node || node away from injection point)* **then**
3         inject new node;
4     **end**
5     **for** *each node $n_i$* **do**
6          Add Gaussian noise to neighbor nodes;
7          Generate Voronoi region;
8          **if** $n_i \in \mathcal{S}$ **then**
9             $L_i \leftarrow$ Algorithm 1;
10           move $n_i$ towards $L_i$;
11           $shift_i \leftarrow d(n_i, L_i)$;
12          **else**
13             Calculate Voronoi Centroid $C_i$;
14             move $n_i$ towards $C_i$;
15             $shift_i \leftarrow d(n_i, C_i)$;
16          **end**
17     **end**
18     **end**
19     *stability* $\leftarrow \sum^N shift_i$;
20     **if** PAC (cf. Equation (2) on page 12) $\geq$ *minCoverage* **then**
21         *count* = *count* + 1;
22         *minCoverage* $\leftarrow$ PAC;
23     **end**
24 **end**

---

The objective function for each candidate solution $n_i'$ (used in line 11 of Algorithm 1):

$$\text{obj}_{fun}(n_i') = \begin{cases} \frac{A_{V_i}}{A_{S_i}} \times \frac{1}{d(n_i, n_i')} & \text{if } N_{n_i'} \geq N_{n_i} \\ 0 & \text{otherwise} \end{cases} \tag{1}$$

where

- $A_{V_i}$ (used line 9 in Algorithm 1) is the node's Voronoi area;
- $A_{S_i}$ (used in line 10 in Algorithm 1) is the node's sensing range area;
- $d(n_i, n_i')$ is the distance between the new candidate ($n_i'$) and the existing ($n_i$) location;
- $N_{n_i'}$ is the number of neighbors at the new location;
- $N_{n_i}$ is the number of neighbors at the existing location.

See Figure 1 for a visual explanation of the parameters used.

*3.2. The Genetic Algorithm—Basic Algorithm*

The basic algorithm for the GA (provided as Algorithm 1 on page 8, above) is explained below:

1. **Initialization**: Initially, *pop_size* (the GA population size) candidate solutions (locations) inside the respective Voronoi region are generated randomly (line 3 of Algorithm 1 on page 8) and

evaluated using the objective function (line 11). Based upon this, the best $\frac{pop\_size}{2}$ solutions are kept as parents for the next GA generation (line 13).

2. **Genetic Operators**: Two standard operators, *cross-over* (lines 15–17) and *mutation* (line 26) are used. The crossover step is implemented as a single-point crossover: two parent solutions are chosen at random and their y-axis values are switched, thereby generating two new offsprings. This process is repeated $\frac{pop\_size}{2\times2}$ times (because parent and offspring generation are of the same size). To ensure diversity, mutation is applied to the offspring generation.

3. **Selection**: the best $\frac{pop\_size}{2}$ from the parent and offspring generation are kept as the next generation (line 27); the best member of this generation becomes the new sensor position.

---

**Algorithm 3: GA-BISON (*conditional*).** The algorithm takes the following input: *minNeighbours* (a threshold value below which the GA is used, set to 4), *minCoverage* (a threshold for minimum coverage to be achieved before the algorithm is used, initially set to 50), *stopCDT* (a termination criterion on the observed node movement, set to $\frac{R_S}{100}$) and *stopCount* (a second termination criterion on iterative coverage improvements, set to 15).

---

1  **while** *(stability > termination|| count < stopCount)* **do**
2   **if** *(exist disconnected node || node away from injection point)* **then**
3    | inject new node;
4   **end**
5   **for** *each node $n_i$* **do**
6    Find number of neighbors $N_i$;
7    Add Gaussian noise to neighbor nodes $N_i$;
8    Generate Voronoi region;
9    **if** *for $n_i$ ($0 < N_i < minNeighbours$)* **then**
10    | $L_i \leftarrow$ Algorithm 1;
11    | move $n_i$ towards $L_i$;
12    | $shift_i \leftarrow d(n_i, L_i)$;
13   **end**
14   **else if** *$N_i \geq minNeighbours$* **then**
15    | Calculate Voronoi Centroid $C_i$;
16    | move $n_i$ towards $C_i$;
17    | $shift_i \leftarrow d(n_i, C_i)$;
18   **end**
19   **else**
20    | $n_i \leftarrow n_i$;
21    | $shift_i \leftarrow 0$;
22   **end**
23  **end**
24  *stability* $\leftarrow \sum^N shift_i$;
25  **if** PAC (cf. Equation (2) on page 12) $\geq minCoverage$ **then**
26   $count = count + 1$;
27   $minCoverage \leftarrow$ PAC;
28  **end**
29 **end**

---

### 3.3. The Genetic Algorithm—GA-BISON (Fixed Nodes)

As the name suggests, in this variation of BISON the genetic algorithm is applied to a fixed number of nodes (we experimented with using 1, 3, 7 or 10 in our performance evaluation, cf. Section 4.3 or Figure 5). Furthermore, as visualized in Figure 7, the genetic component of the approach is only used

after at least 50% coverage has been achieved (as there is very limited benefit in doing so from the start). This is the tunable parameter *minCoverage* in Algorithm 2.

The implementation strategy of GA-BISON fixed nodes is to randomly choose drones from the existing swarm who will apply the GA to determine their next target location to move to (while the remaining drones will continue to use the original BISON algorithm). The pseudo-code for the GA-BISON (fixed nodes) algorithm is provided as Algorithm 2 on page 10.

### 3.4. The Genetic Algorithm—GA-BISON (Conditional)

The conditional variation of the GA approach is used from the beginning but applied only to nodes with a small number of neighbors (three or fewer). The reasoning is that nodes which are surrounded by neighbors are unlikely to improve the WSN performance by making large movements (on the contrary, at this stage they are optimizing the WSN locally through small position changes). The pseudo-code for the GA-BISON conditional algorithm is provided as Algorithm 3 on page 11.

## 4. Setup and Performance Measures

### 4.1. Simulation Setup

To compare the performances of the algorithms, we simulated an environment (square area) into which a drone swarm was deployed to realize a WSN. As shown in Figure 3, we considered obstacle free spaces and spaces with objects scattered though the environment. In addition, we investigated how the approaches could handle communication noise. Obstacles block the movement of nodes; communication noise affects how accurately a node can determine the position of neighboring nodes, used to calculate the node's Voronoi region. We investigated whether adding a moderate Gaussian noise effect to the communication (noise standard deviation of 0.05, [54,55]) significantly impacted system behavior. The simulation terminates/stops when either (a) the average shift of the nodes drops below a threshold value (see below) or (b) the overall area coverage no longer improves.

For all simulations, the nodes' sensing range was assumed to be $R_S = 10$ m (meters); the communication range (defined relative to $R_S$) was $R_C = \sqrt{3}R_S$. The environment into which the swarm was deployed was simulated to be 100m×100m. Drones could only enter this environment at a dedicated *injection point*, situated at position (0,0). The threshold value for the minimum average node shift, used as a termination criterion for the algorithm, was set to $\frac{R_S}{25}$ (for more details on any of these settings/the motivation, we refer to [54,55]). The simulations were implemented in MATLAB R2017b.

### 4.2. Evaluation Metrics

Following the argumentation in earlier work, and in accordance with the literature, performance of the algorithms is measured through three metrics (cf. [54,55]). The three metrics used are:

- The **number of simulation steps** required to achieve optimum performance (full coverage). This is a straight forward metric to measure the speed with which the algorithms converge on the optimal solution. Since the execution of the algorithms is happening on each device separately and under the assumption that the actual flying operation takes more time than the calculations, comparing the performance on the basis of the simulation steps required to achieve full coverage enables us to relate the approaches to one another with regard to deployment speed.

- The **percentage area coverage (PAC):** following from the previous metric, the coverage achieved is the primary measure for performance. By considering not only the moment when this is fully achieved, but also its development over time until then, we can compare the approaches progress during execution. For the measure PAC, the coverage of 100,000 sampling points inside the environment into which the swarm is released is recorded. Given these, PAC is calculated:

$$\text{PAC} = \frac{Number\ of\ covered\ points}{Total\ number\ of\ (accessible)\ points} \times 100 \qquad (2)$$

The sampling points are drawn from the accessible area inside the environment, meaning that pillars/signal scattering objects are excluded. This is of relative little impact here, but for investigations on environments with a larger number of walls and objects (not included in this report) this makes a significant difference. The area coverage is expressed as a percentage precisely because this allows us to compare the results from different environments.

- The **cumulative distance traveled (**CDT**):** after considering the steps to completion and the evolution of coverage; until then we use CDT as a measure for comparing the physical effort required (i.e., the cost) to do so. CDT is calculated as the sum of all node-movements over time:

$$\text{CDT}(n) = \sum_{i=\{1,\dots,n\}} cnm(t_i) \tag{3}$$

with *cnm* the sum of the combined node movements during a time step, defined recursively as:

$$cnm(t) = \begin{cases} \frac{1}{N(t)} \sum_{i \in N(t)} |d(L_{n_i}^{t-1}, L^t)| & t > 1 \\ \frac{1}{N(t)} \sum_{i \in N(t)} |d(L_{n_i}^0, L^t)| & t = 1 \end{cases}$$

with $|d(L_{n_i}^{t-1}, L^t)|$ the absolute Euclidean distance between the previous location of node $n_i$ and its current one, where $L_{n_i}^{t_0}$ is the injection point at time $t_0$.

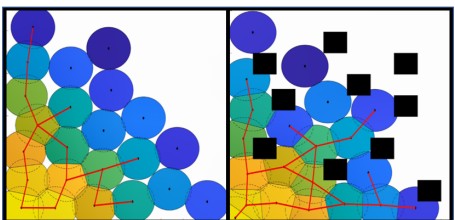

**Two types of environment: obstacle-free (left) and with "scatterrers", i.e. obstacles that block node movement (e.g., load-bearing pillars).**

**Figure 3.** Two different types of indoor environments were simulated: one where the swarm of drones was dispatched into one large indoor environment (e.g., a large factory hall) and another wherein the indoor environment was filled with obstacles (such as, e.g., pillars). In the images above, the red lines indicate drones connected by proximity. In both cases there are drones at the outer edge of the swarm that are temporarily not connected. As explained in Section 3.1.1, this is remedied as more drones are added to fill the entire area (see Figure 6 on page 16 for snapshots of the process over time). Scatterers block the passage of drones and make the problem more difficult (and the scenario more realistic).

### 4.3. GA Parameter Settings

Using the above performance measures, a preliminary parameter space exploration was conducted to determine adequate parameter settings for the GA. Specifically, the population size and the mutation rate needed to be fixed, along with the number of generations (of candidate solutions) produced (i.e, the number of iterations before picking location to move to). Therefore, these three parameters were investigated separately using PAC, CDT and *steps* as objective functions. Our parameter space exploration was undertaken for the BISON *fixed nodes* algorithm, where results were generated for 1, 3, 7 and 10 nodes. Below we report on these individual investigations for population size, number of generations and mutation rate, respectively; the results are shown in the overview in Table 4.

**Table 4.** Results for the parameter-space exploration for obstacle-free environments without communication noise, using the GA-BISON fixed nodes algorithm. Shown are: the percentage of the area covered `PAC`, the cumulative distance travelled by all nodes `CDT` as well as the number of steps taken **st**) for four different numbers of fixed nodes (1,3,7,10). For each parameters *population size* (**p**), *# iterations* (**#**) and *mutation rate* (**m**) three values were investigated in isolation. The highlighted values were used for the performance evaluation provided in Section 5. Figure 4 shows this graphically.

| | GA-BISON 1 nodes | | | GA-BISON 3 nodes | | | GA-BISON 7 nodes | | | GA-BISON 10 nodes | | |
|---|---|---|---|---|---|---|---|---|---|---|---|---|
| | PAC (%) | CDT (m) | st | PAC (%) | CDT (m) | st | PAC (%) | CDT (m) | st | PAC (%) | CDT (m) | st |
| **p** = 10 | 93.68 | 15.8 | 426 | 94.00 | 50.03 | 467 | 82.68 | 81.15 | 307 | 85.56 | 71.16 | 341 |
| ± | 1.2 | 2.69 | 48.7 | 3.4 | 5.1 | 68 | 2.2 | 12.1 | 53 | 3.13 | 26 | 63 |
| **p** = 15 | 95.78 | 22.7 | 526 | 84.64 | 41.06 | 330 | 85.42 | 108.5 | 440 | 80.27 | 147.6 | 454 |
| ± | 1.4 | 3.27 | 54.1 | 2.3 | 4.2 | 52 | 3.1 | 14.8 | 62 | 2.05 | 32 | 71 |
| **p** = 20 | 96.87 | 23.62 | 545 | 90.42 | 56.82 | 467 | 80.84 | 81.17 | 293 | 80.9 | 121.6 | 316 |
| ± | 1.9 | 4.06 | 56.3 | 2.9 | 5.5 | 63 | 1.6 | 12.1 | 48 | 2.21 | 28 | 56 |
| **#** = 15 | 96.87 | 23.14 | 545 | 90.42 | 56.82 | 467 | 80.84 | 81.17 | 293 | 80.9 | 121.6 | 316 |
| ± | 1.7 | 2.5 | 49 | 4.5 | 14.6 | 77 | 2.58 | 15.9 | 28 | 0.61 | 27 | 16 |
| **#** = 30 | 93.8 | 16.15 | 418 | 80.92 | 23.75 | 273 | 87.98 | 99.74 | 397 | 82.15 | 59.69 | 273 |
| ± | 1.2 | 1.6 | 42 | 4.1 | 9.6 | 58 | 3.47 | 17.8 | 36 | 1.11 | 13 | 14 |
| **#** = 50 | 98.2 | 18.9 | 600 | 92.67 | 56.31 | 532 | 83.3 | 54.65 | 351 | 82.41 | 63.99 | 301 |
| ± | 1.7 | 1.8 | 70 | 4.7 | 14.4 | 103 | 2.92 | 13.7 | 31 | 1.24 | 16 | 15 |
| **m** = 0.05 | 96.87 | 23.67 | 545 | 90.42 | 56.82 | 467 | 80.84 | 81.17 | 293 | 80.9 | 121.6 | 316 |
| ± | 1.8 | 1.5 | 71 | 1.2 | 3.65 | 19 | 1.06 | 9.6 | 30 | 1.16 | 17 | 34 |
| **m** = 0.1 | 93.6 | 20.61 | 429 | 92.95 | 52.66 | 489 | 83.6 | 71.4 | 272 | 83.59 | 129.8 | 359 |
| ± | 1.5 | 1.2 | 66 | 1.3 | 2.9 | 23 | 2.04 | 7.4 | 28 | 1.94 | 19 | 37 |
| **m** = 0.2 | 92.69 | 20.09 | 399 | 93.14 | 55.99 | 530 | 81.58 | 97.84 | 351 | 81.06 | 87.98 | 421 |
| ± | 1.3 | 1.2 | 54 | 1.4 | 3.61 | 26 | 1.49 | 10.4 | 33 | 1.57 | 14 | 39 |

The above table presents the results for independent evaluation of the three parameters *population size* (**p**), *# iterations* (**#**) and *mutation rate* (**m**). These choices for the respective missing parameters were initially guided in part by the existing values in literature (e.g., for the mutation rate) and in part by the design of the model problems we aim to solve (such as population size), but later confirmed by the results obtained. The values of parameters outside the ranges shown in the Table do not show promise of better overall performance (with respect to `PAC`, `CDT` and steps).

It is likely that the changes in these parameters would impact performance similarly across algorithms. Furthermore, the evaluated scenarios are simplifications and we fully expect to have to perform additional analysis before deploying drones in the wild. Since the parameters for specific real world scenarios will very likely be scenario dependent, we argue that the chosen ones suffice for our evaluation here and we hypothesize that they are good guesses for the initial iteration.

### 4.3.1. Population Size (Figure 4, top row)

When fixing the population size, we can notice that when the number of nodes applying the GA increases, the number of simulation steps and the percentage area coverage decrease, while the cumulative distance traveled increases (cf. Figure 4). This is because with a greater number of nodes applying the GA comes larger movements to discover the region of interest in one simulation step.

Furthermore, in Figure 4 we can also notice that for lower number of nodes, the increase in the population size increases the number of steps; while for a larger number of nodes, higher population sizes yield in fewer steps with a slight reduction in `CDT` values. In our implementation, we care more about discovering more possible solutions when implementing a GA, to have a better and

wider discovery process. Therefore, we have chosen population size 20 to be used during GA-BISON implementation, to make use of its benefits on PAC and number of steps.

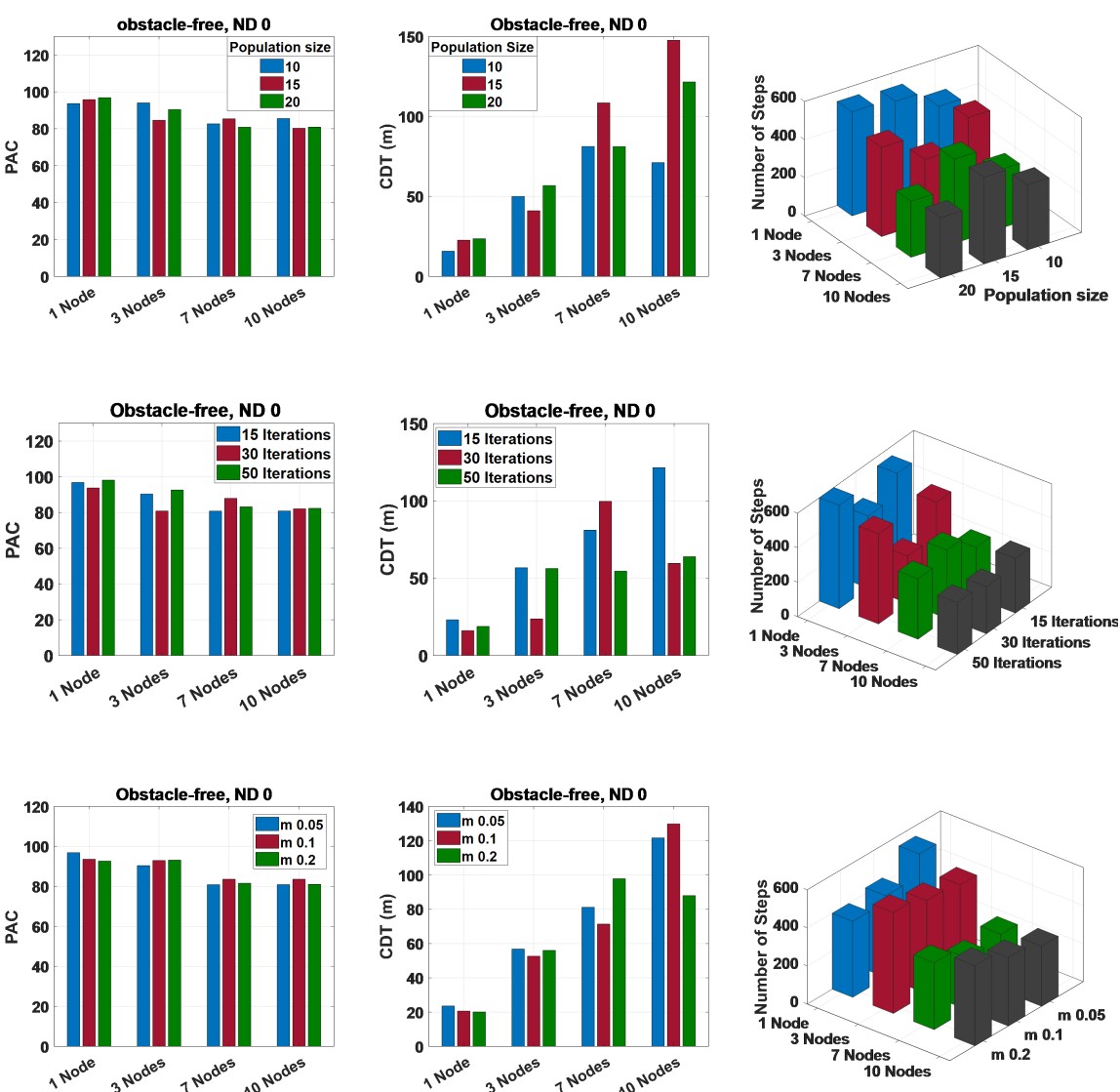

**Figure 4.** Results for the parameter space exploration for the population size for the genetic algorithm. Reported are PAC (left panels), CDT (middle panels) and generations of candidate solutions (iterations, right panels). The numerical values are reported in Table 4, cf Section 4.3 for a discussion.
**Top row:** Parameter space exploration for **population size** (using 15 iterations and a mutation rate of 0.05) in obstacle free environments without communication noise. Given these results, the value *population size* = 20 was used. **Middle row:** Parameter space exploration for **number of iterations** (using a population size of 20 and a mutation rate of 0.05) in obstacle free environments without noise. Given these results, *number of iterations* = 15 was used. **Bottom row:** Parameter space exploration for the **mutation rate** (using 15 iterations and a population size of 20) in obstacle free environments without communication noise. Given these results, 0.05 was used for the *mutation rate*.

### 4.3.2. Number of Iterations (Figure 4, middle row)

Considering the final coverage achieved (Figure 4, middle row, left panel) we find that the number of iterations doesn't significantly alter the outcome, once we use at least 15 iterations. Therefore, to reduce the energy consumption, simulation time, and computation complexity, we chose 15 iterations to be processed in the system. We acknowledge that, for seven fixed nodes (the variation of BISON *fixed nodes* used for our noise coherence analysis (Section 5.2, Figures 12 and 13), 30 iterations would

result in marginally better WSN coverage (Figure 4, middle row, left panel); however, as this comes at a greatly increased cost (Figure 4, middle row and panel) we argue that 15 is an appropriate choice.

### 4.3.3. Mutation Rate (Figure 4, bottom row)

The results' dependence on the mutation rate affects the number of required steps and the `CDT` values, in such a way that higher mutation rates increase the number of simulation steps and `CDT` (cf. Figure 4, bottom row). The percentage area coverage (`PAC`) is not significantly affected by the change in the mutation rate, while the increase in the number of nodes reduces the percentage area coverage achieved in the system when fixing the mutation rate. On balance, a mutation rate of 0.05 provides a moderate level of `CDT` with the least number of simulation steps.

### 4.4. The Number of Fixed Nodes in BISON-GA (Fixed Nodes)

As shown in Figure 5a, in the obstacle-free environment, increasing the number of affected nodes improves `PAC` until seven nodes are used. The improvement in performance with regard to `PAC` and *steps* (cf. Figure 5a,c) seen when increasing the GA-enhanced nodes to seven is less pronounced when there are obstacles present in the environment. Due to this, Section 5.2, which is dedicated to our separate investigation on noise coherence analysis, compares results for an obstacle-free environment, that were generated using seven fixed nodes in the *fixed nodes* algorithm. To motivate this choice further, the investigations on the overall performance of the algorithms, Section 5.1, compare results for 1, 3, 7 and 10 nodes (cf. Figures 8–11) and contrast the performance in an obstacle-free environment (cf. Figures 6, 8 and 9) with the performance in an environment with obstacles (cf. Figures 10 and 11).

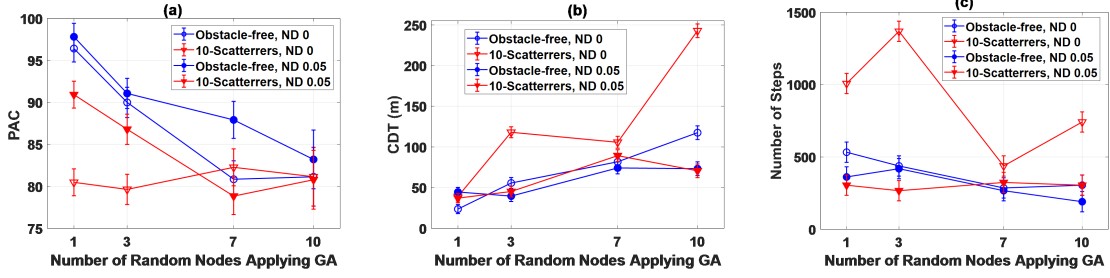

**Figure 5.** Performance comparison over the number of fixed nodes (x-axis) in the GA-BISON *fixed nodes* algorithm for both noise settings. Plotted are `PAC` (**a**), `CDT` (**b**) and number of simulation steps (**c**) (cf. Section 4.2 for details on these three).

In contrast to what we observe for `PAC` and *steps*, `CDT` performance (cf. Figure 5b) decreases almost always when the number of nodes using the GA is increased.

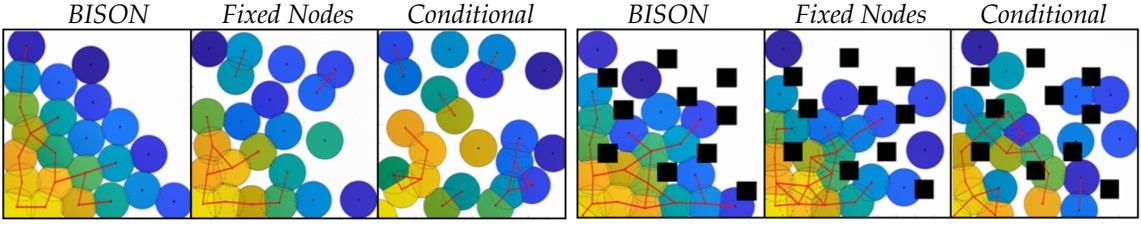

**Figure 6.** A snapshot showing—for both scenarios—an intermediate stage of coverage using BISON (left), GA-BISON *fixed nodes* (middle) and GA-BISON *conditional* (right). Shown are covered areas per drone/node (circles) and inter-node connections (red lines). These images are provided here to illustrate the difference in swarm behavior during deployment; for numerical comparison, see Figures 7–11 (algorithm performance, measured in `PAC`, `CDT` and steps, defined by Equations (2) and (3), respectively) and Figures 8–11 (drone movement, expressed as diffusion (*D*) and drift (*F*), defined by Equations (4) and (5), respectively).

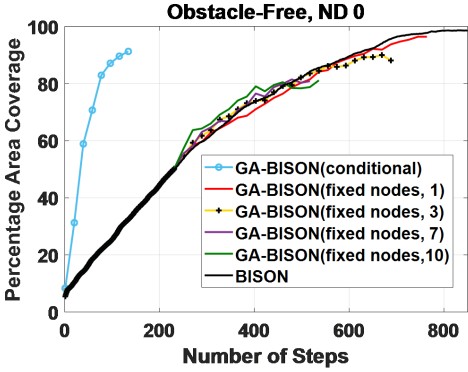
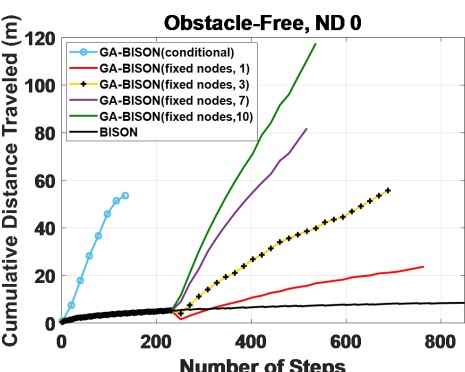

**Figure 7.** Performance analysis of the algorithms; plotted (y-axis) are `PAC` (left) and `CDT` (right), over the number of steps (x-axis). The shown results are for obstacle-free environments and without added noise. As discussed in Section 3.3, the BISON *fixed node* algorithm starts using the GA *after* at least 50% coverage has been achieved. Therefore the initial trajectory of the respective graphs (shown here in black) is that of the original BISON, with the graphs for the different numbers of fixed nodes diverging once BISON has achieved 50% coverage. For readability, this is omitted in Figures 8–11 by starting the plots at the 50% mark on the y-axis (for `PAC`) or at the 0 mark (for `CDT`).

As already mentioned above, it also turns out that—maybe surprisingly—adding noise improves performance, and happens with both `CDT` and the number of required steps (Figure 5(c)), but it does so differently depending on the numbers of nodes to which the algorithm is applied to.

## 5. Results and Discussion

We first (Section 5.1) evaluate algorithm performance (summing up our findings in Table 5 on page 20) and then (Section 5.2) investigate the noise coherence properties of the WSN drone swarm. For the performance analysis we consider both obstacle-free and obstacle-rich environments separately (Sections 5.1.1 and 5.1.2, respectively) before comparing them in Section 5.1.3. We show the impact of noisy communication on algorithm performance (Section 5.1.4) and conclude with a summary.

Up to this point, our investigations were primarily theoretical in nature. Section 5.2 focuses on the drift and diffusion of the members of a swarm under noisy communication channels. Combined with the insight that some noise actually improves swarm performance, this servers as a motivation to continue our work, with the testing of the approaches using real drones as a goal.

### 5.1. Performance Evaluation of the Algorithms

5.1.1. Performance Evaluation—Obstacle-Free Environment

We measure the percentage of the covered area (`PAC`), plotted in Figure 8; for numerical results of the fixed node approach see Table 4. The final performances are comparable across algorithms but the original BISON approach takes ≈850 steps to terminate. Notably, we can clearly see when comparing the left and the right panel of Figure 8 that with regard to `PAC`, adding noise improves the performances across the board. With regard to the discovery rate, that is, the evolution of `PAC` over time, the plots in Figure 8 show both a steeper gradient and a higher final result when noise is added. This means that in noisy environments, the algorithms perform better faster. The `CDT` allows comparison of the operational cost incurred by the algorithms. The graphs in Figure 9 clearly show that the original approach absolutely outperforms both of the proposed algorithms using the GA. While the original BISON continued to run for an additional 450 steps before terminating, the `PAC` achieved after 400 steps is already very good (cf. Figure 8) while incurring a fraction of the movement cost of the enhanced algorithms (cf. Figure 9). Regarding algorithm efficiency, the conditional approach terminates after around 150 iterations, while fixed node takes significantly longer but still less than the original approach (which requires ≈850 steps to terminate).

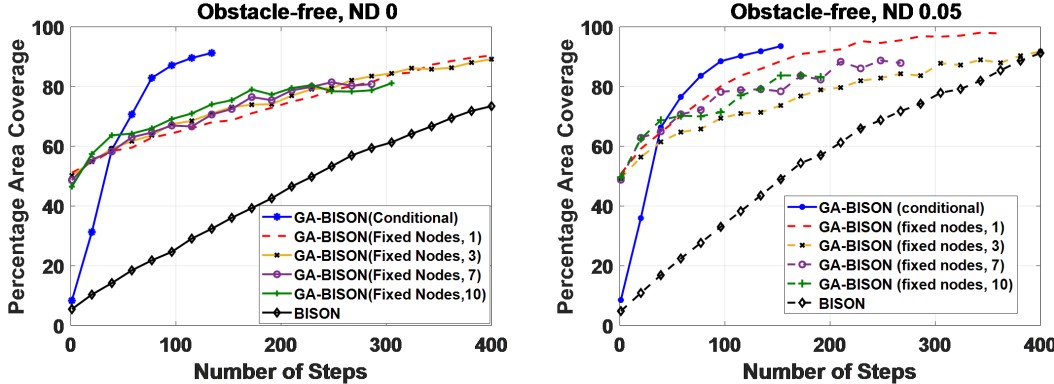

**Figure 8.** `PAC` (y-axis) plotted over the number of steps (x-axis). We compare performances in an obstacle-free environment for perfect communication (**left**) and (**right**) when communication is subjected to noise. Note: the *fixed nodes* algorithm only starts after 50% coverage is achieved, which is why the respective graphs start at this mark on the y-axis; cf. Figure 7.

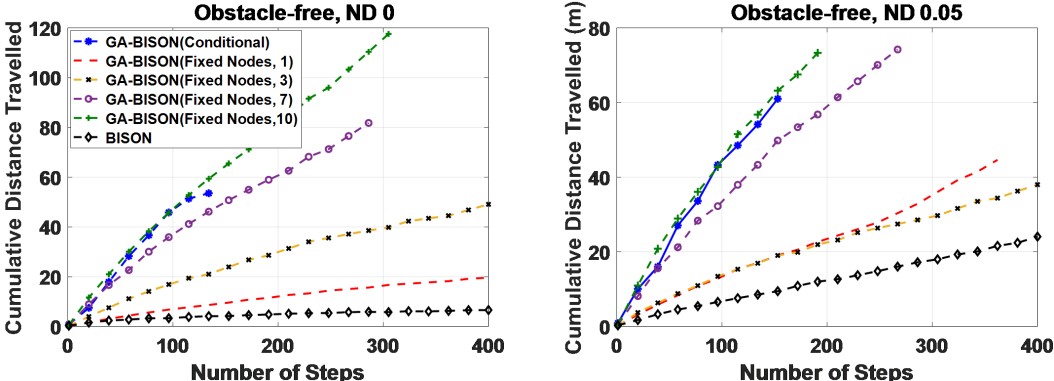

**Figure 9.** `CDT` (y-axis) plotted over the number of steps (x-axis). The plots compare the algorithms in an obstacle-free environment for perfect communication (**left panel**) and (**right panel**) when communication is subjected to a Gaussian noise effect (standard deviation: 0.05).

### 5.1.2. Performance Evaluation—Obstacle-rich Environment

As before, WSN coverage is given by the achieved final `PAC` in Figure 10 while the evolution thereof over time is the discovery (the steeper the better). Again, all approaches perform similarly, with conditional approach marginally outperforming the others, and as before, the original BISON approach taking ≈850 steps. As in the obstacle-free environment, the three algorithms show distinctly different performance behaviors with all variations of the fixed node approach performing almost the same. Again, adding noise (Figure 10, right panel) improves performances. Regarding the discovery rate, the conditional approach quickly gains coverage, slowing down only towards final performance values. In contrast, the other approaches' discovery rate only increases (almost) linearly.

As before, BISON continues to outperform the GA approaches when obstacles are added to the environment (shown as `CDT` in Figure 11). Regarding the efficiency, the conditional approach terminates around 200 iterations; fixed node takes significantly longer with noise improving on this significantly though. The original BISON again performs worst of all.

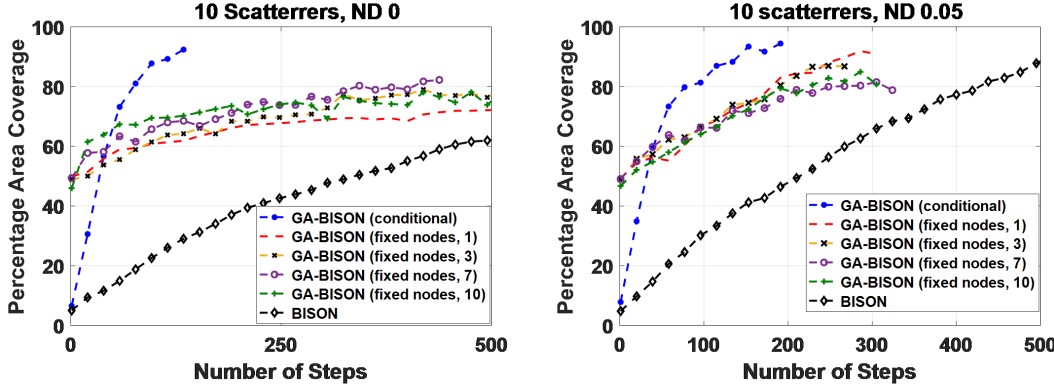

**Figure 10.** `PAC` (y-axis) plotted over the number of steps (x-axis). The plots compare the `PAC` performance of the algorithms in an environment with obstacles for perfect communication (**left panel**) and (**right panel**) when communication is subjected to a Gaussian noise effect (standard deviation: 0.05). Note that the *fixed nodes* algorithm only starts when 50% coverage have been achieved, which is why the respective graphs start at this mark on the y-axis; cf. Figure 7.

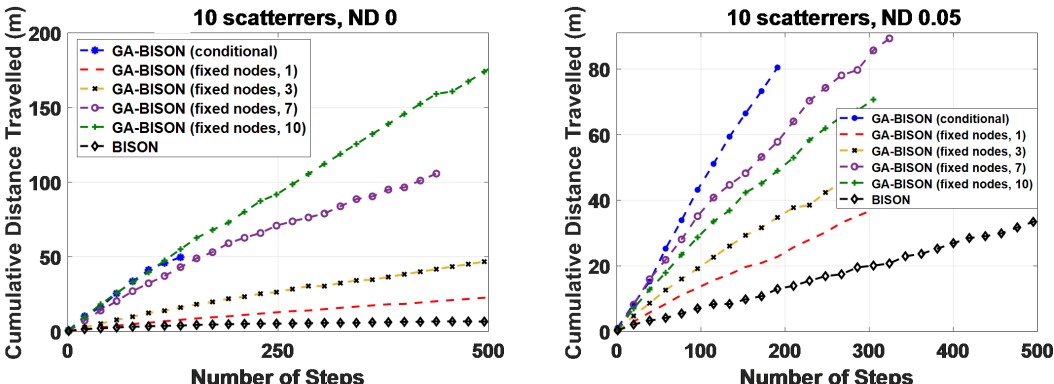

**Figure 11.** `CDT` (y-axis) plotted over the number of steps (x-axis). The plots compare the `CDT` performances of the algorithms in an environment with obstacles for perfect communication (**left**) and (**right**) when communication is subjected to a Gaussian noise effect (standard deviation: 0.05).

### 5.1.3. Obstacle-free versus Obstacle-rich Environment

We compare the impact of different environments on the performance the algorithms, i.e., the results for an obstacle-free environment (Figures 9 and 8) with those obtained in the obstacle-rich environment (Figures 11 and 10). As the individual results are already discussed in the previous sections we focus on the environment related insights here only.

When comparing the `PAC` in Figure 8 with those in Figure 10, the similarities are striking, both in the absence (respective left panels) and the presence (respective right panels) of noise. Note that in Figure 10, the x-axis shows the first 500 steps, 25% more then the plots in Figure 8. This 25% increase is needed because adding objects affects performance consistently for both measures (`PAC`, `CDT`) and in both environments. Adding objects or noise also improves the discovery rate, significantly. Also in both environments With regard to the aggregated nodes' movement (i.e., the cost for the WSN deployment), the original BISON achieves significantly better coverage with a lot less movement in either environment. As for the efficiency of the approaches, it turns out that interestingly, adding objects results in longer algorithm run times but adding noise has a detrimental effect in the obstacle-free environment while it actually improves algorithm run-time in the presence of obstacles.

### 5.1.4. The Impact of Noise

We investigate the impact of adding noise to the communication, i.e., the impact of imprecision in the centroid calculations. As motivated in Section 5.2.1, estimating the neighboring nodes distances from the time of arrival of the transmitted signal is always inaccurate in real implementations. This is due to both internal and external factors, including the device performance, and the outside interference with other signals. Therefore, our contribution in this work is including the effect of noise as a randomization technique that can simulate the inaccuracy of determining the localization of the neighboring nodes distances. In contrast to the approach in [68], where randomization was utilized, we wanted to examine a clearly defined noise function. Having several factors affecting a certain measurement will tend the overall behavior of the random numbers to follow normal distribution. In addition to that, since the environment we are dealing with is unknown, it was reasonable to consider the general form of noise distribution which is the Gaussian distribution [94,95]. Our analysis is concerned with distributing the mobile nodes inside a closed region of interest; hence our consideration of noise is limited to indoor communication noise. An interesting note mentioned in [96,97] is that even if the fading noise mechanisms are characterized by multiples of Gaussian distribution or other distribution forms, the overall contribution of these mechanisms will eventually tend toward Gaussian distribution. Therefore, we used the simplest model of Gaussian distribution with zero mean to describe the noise affecting the sensing technique used to find the distance between the neighboring sensors. We can compare the impact of noise by comparing the left panels (no noise) with the right panels (noise added, standard noise deviation: 0.05) in Figures 8–11.

With regard to coverage (Figures 8 and 10), adding small amounts of noise improves the total coverage achieved by the WSN. Similarly, `PAC` increases faster under noise, with larger gains achieved at an earlier deployment stage. However, for BISON and the conditional approach (independent of the number of obstacles) noisy simulations incurs a higher cost (cf. `CDT` in Figures 9 and 11). This also holds for the fixed node approach *except* when only a single node is using the GA.

As far as the algorithm efficiency is concerned, we previously noted that adding objects to the environment always slows the swarm down. This is also the reason for reporting 500 iterations in Figures 11 and 10 (as opposed to 400 in Figures 9 and 8). Furthermore, the steeper gradients observed for the `CDT` plots (representing increased node movement when adding noise) in Figures 9 and 11 clearly indicates that noise also results in the nodes moving faster.

### 5.1.5. Summary and Discussion

Our findings regarding the performances of the algorithms are summed up in Table 5. Given the fact that energy can be expected to be a limited resource for many WSN and certainly for most mobile WSN, the energy demands of the algorithms might be the most important measure. In that regard the original algorithm clearly outperforms its GA offsprings. This is not surprising as it was designed with simplicity in mind. Given the higher energy cost for the new algorithms it is also logical that the increased cost seems proportional in the number of nodes using the GA.

**Table 5.** A brief summary and interpretation of the performances shown in Figures 5 and 7–11 above: for both **discovery rate** and **coverage**, *fast* or *high* is preferable; for **noise tolerance** *robust* means unaffected; the **energy demands** compare the requirements of the respective algorithms.

| | | Discovery Rate | Coverage | Noise Tolerance | Energy Demands |
|---|---|---|---|---|---|
| **BISON with GA** | (**conditional**) | fast | low to moderate | robust | high |
| | (**fixed nodes**) | low to moderate | moderate | moderate | moderate to high |
| **BISON** | | low | high | high | low |

The positive impact of communication noise on performance is clearly visible across Figures 9–10 (left panels: no noise; right panels: noise). As noise causes more *courageous* exploration it also makes intuitive sense that this is least beneficial for the algorithm where a fixed number of nodes use the GA (as opposed to none or only those at the outer fringe of the network).

The benefits of augmenting BISON with a GA becomes evident when considering the deployment speed where it again is increasingly beneficial to have more nodes use the GA (fixed nodes) and adding this only for those nodes at the outer edge of the network (conditional). However, a high discovery rate comes at the cost of coverage, which is inverse proportional to the deployment speed.

In Section 2.2 we discuss alternatives to using GA, however, this article only provides results for a hybrid approach using BISON and GA. A comparative evaluation against hybrid approaches using other heuristics is outside the scope of this paper. However, prior to embarking on our project, a comparison with the NSVA algorithm [54,98] was performed. We found the original BISON algorithm to be superior in the following ways: in terms of coverage and the number of nodes, BISON tends to achieve maximum coverage at an earlier time and with fewer nodes compared to the NSVA algorithm (with ratio of 1 to 3), under the same parameters.

### *5.2. Noise Coherence Analysis*

As noise significantly impacts WSN deployment properties, a short investigation of the effect it has on node movement is in order. This is further motivated by the desire to apply the theoretical work to practical implementations for swarms of (semi)-autonomously operating drones. With that in mind, the average device velocity and the changes of this value over time offer valuable insights with regard to the expected behavior of a drone swarm using the proposed algorithms to deploy an indoor WSN.

### 5.2.1. Motivation

In any real-world deployment of robotic systems there is the potential for incurring increasing positional errors over time [99,100]. This is a challenge addressed in various ways in the literature. Therefore, with regard to noise and positioning error, this article focuses on the impact of noisy measurements on the positioning of a device (where the positioning itself is expected to be error-less in itself). Seeing the impact of noise on standard network performance parameters motivates the need to understand this in a fundamental, Fokker-Planck formalism, i.e., a formal equation describing the evolution of the probability density function of the velocity of a particle over time. Following the literature [101] and with the scope of the paper in mind we use empirical kinematic data (generated by our simulations) to estimate the diffusion and the drift coefficients of this equation (as opposed to a formally defined theoretical (complex) model/Fokker-Planck equation to determine node-velocity).

### 5.2.2. Modeling Node—Diffusion and—Drift

We refer to the mean rate of change of average node velocity as *diffusion* ($D$) and to the evolution thereof as *drift* ($F$). The coefficients for diffusion ($D$) and drift ($F$) are approximated as follows [101]:

$$D\big(v(t)\big) \approx \frac{1}{2}\left\langle \frac{\big(v(t+\delta t)-v(t)\big)^2}{\delta t} \right\rangle \tag{4}$$

$$F\big(v(t)\big) \approx \left\langle \frac{v(t+\delta t)-v(t)}{\delta t} \right\rangle \tag{5}$$

with

$$v(t) = \frac{1}{n}\sum_{i=1}^{n} v_i(t)$$

where $v(t)$ is the average velocity of all the available nodes $n$ at time $t$, $D\big(v(t)\big)$ is the diffusion coefficient, $F\big(v(t)\big)$ the drift coefficient and $\delta t$ the chosen time step value.

### 5.2.3. The impact of noise

Figures 12 and 13 show the outcome of our investigations into the impact of noise. Plotted are diffusion (*D*) and drift (*F*), calculated using empirical data from simulations as opposed to using a formally defined theoretical (complex) model/equation to determine node-velocity). Since the only variable difference between the respective left (blue) and right (red) panels is the noise-level (*no noise on the left in blue, noise-level ND = 0.05 on the right in red*), it is clear that the motion of the sensors is affected by noise. The top rows in both panels shows diffusion and drift for GA-BISON *fixed nodes* with seven fixed nodes. The motivation for reporting the results for seven nodes is given in Section 4.4.

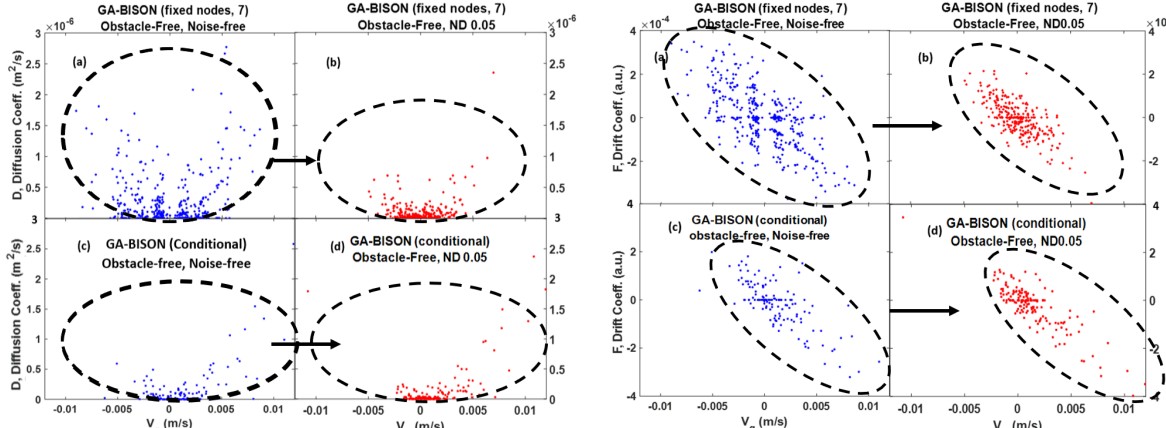

**Figure 12.** Noise coherence analysis: shown are (**left panel**) *diffusion* (Equation (4)) and (**right panel**) *drift* (Equation (5)). The respective top plots show the impact of noise (arrow) incurred by the BISON *fixed nodes* algorithm; the respective bottom plots show the same for BISON *conditional*. In the previous section we discussed algorithm performance and noted that BISON *conditional* was the least affected by noise (cf. `PAC` (Equation (2)) in Figures 8 and 10); this is reflected by the lower plots in both panels above, where hardly any change can be observed in either node—diffusion or drift. In contrast, as seen in Figures 9 and 11, when subjected to noise BISON *fixed nodes* showed a reduction in `CDT` (Equation (3)) and steps, which is visible in the reduced drift and diffusion of the swarm, shown in the upper plots of both panels.

### The Impact of Noise in General

The results in Figures 8–13 clearly show the benefit of noise for the resulting final WSN coverage and the rapid increase thereof during swarm deployment. The most likely quantitative explanation is that the presence of noise (of a certain kind and under the right circumstances, etc.) leads to different directional switching that can be considered helpful in accomplishing some WSN tasks. This is comparable to the positive impact of noise in biological systems [101].

### The impact of noise: Fixed Nodes versus Conditional

When comparing the two BISON-GA approaches *fixed nodes* and *conditional* as we do in Figures 12 and 13 we see that, under the same noise level the values for the fixed node approach have a wider distribution than for the conditional approach. This finding is an important indication for the results revealed previously in Figures 8 and 11, where GA-BISON conditional was least affected by the presence of noise. On the other hand, as the noise level increases, we observed an enhancement in GA-BISON fixed nodes in terms of reducing the `CDT` values, which is a reflection of the reduction in the random motion of nodes (Diffusion, shown in the left panels of Figures 12 and 13) and their change in their average velocities (Drift, shown in the right panels of Figures 12 and 13).

### 5.2.4. Summary

Neighbourhood information plays an important role [102]. The quality of this information is impacted by the noise in the system and we discussed the impact of adding noise using the results plotted in Figures 8–13. When comparing the impact of noise (blue, no noise, versus red, with noise) for either Diffusion or Drift we see that the impact is more pronounced for the fixed node approach but still noticeable for the conditional approach. This means that the presence of noise, which is a likely occurrence in a system deployed in the real world, is not necessarily a problem. More so, for small amounts of noise the impact can be considered beneficial.

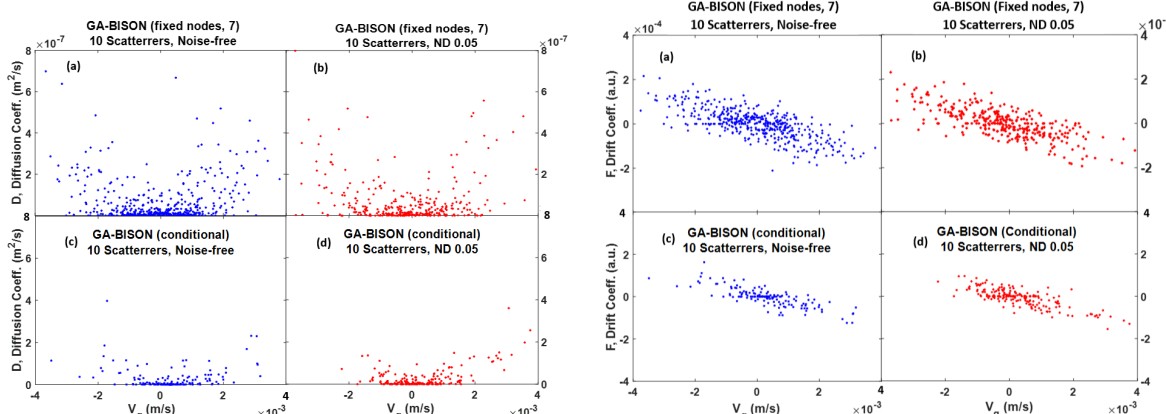

**Figure 13.** (**left panel**) Node diffusion in an obstacle-rich environment (compare this to the left panel of Figure 12, comparing BISON *fixed nodes* (top plots) to BISON *conditional*). Plotted is the diffusion coefficient $D$ (y-axis) defined in Equation (4) versus the velocity (x-axis). Compare these results to Figure 10, where we see that the `PAC` of BISON *fixed node* improves significantly under noise (which is reflected in the reduced diffusion shown in the top plots here) while that of BISON *conditional* hardly shows any improvement (matched by the little change between the two lower plots here). (**right panel**) The drift of the WSN drone swarm in an obstacle-rich environment (compare this to the right panel in Figure 12 for obstacle-free environments), comparing the BISON *fixed node* approach (top plots) to the BISON *conditional* approach. Plotted is the drift coefficient $F$ (y-axis) defined in Equation (5) versus the velocity (x-axis). Compare these results to Figure 11 from the performance analysis, where we see that the `CDT` of BISON *fixed node* is reduced significantly under noise (which is reflected in the reduced diffusion shown in the top plots here) while that of BISON *conditional* remains virtually unchanged (which we see here in lack of change in the lower plots).

## 6. Conclusions and Future Work Plan

We studied the contribution of merging two Nature-inspired algorithms on the performance metrics of WSN deployment and reallocation in unfamiliar regions of interest. The GA was preferred among other algorithms to be implemented with our previously developed BISON algorithm due to its efficiency in large scale applications for maximizing the lifetime of the network, its flexibility of being executed locally on each agent, and its objective function that can be modified by operators.

We introduced two new strategies for implementing a GA with BISON: BISON *fixed nodes*, where random nodes are selected among the available agents at the intermediate deployment stage to apply the GA for their reallocation process; and BISON *conditional*, where each node checks its number of neighbors to decide whether to to make use of the GA to determine its next position. Simulation results revealed that by discovering further locations (as opposed to the Voronoi centroids) both GA-BISON approaches improve the execution time and discovery rate of the network.

However, such implementations require a tradeoff between coverage and the energy consumption of the network. Higher coverage requires more movement from sensor nodes which increases the distance traveled and the overall energy consumption of the network. In contrast, reducing the distance traveled does not always guarantee better coverage or better energy consumption.

We demonstrated that GA-BISON conditional has the fastest discovery rate but with highest energy consumption, followed by GA-BISON fixed nodes with moderate performance in coverage and energy consumption; while BISON approach has the highest regional coverage with least distance traveled but with slowest discovery rate. We also validated that GA-BISON conditional coverage performance is robust against noise effect, while BISON and GA-BISON fixed nodes performances are enhanced but all have a downside on the energy consumption of the network. From these analyses, the efficiency that we can guarantee from the developed approaches depends on the application requirements and abilities, which are summarized in Table 5.

We discussed the change of the GA parameters (population size, number of iterations, and mutation rate) on the performance of GA-BISON fixed nodes,to provide an analytical demonstration on the choice of these parameters that contribute to the efficiency of the GA approach. The population size showed an influence on PAC, CDT, and number of simulation steps required, such that the increase in the population size reduces the PAC and the number of steps but increases the CDT. The change in the number of iterations also reduces the number of steps and increases the CDT, but has a negligible effect on the PAC achieved by the network. Similar behavior is adhered when changing the mutation rate, but with a reduction in the number of steps with an increase in CDT.

Throughout the article references are made to potential future work. With regard to a comparative performance evaluation, pitting GA-BISON against e.g., PSO-BISON or ACO-BISON (discussed in Sections 1.1 and 5.1.5), this will constitute a significant effort, worthy of a separate project and investigation of themselves. We would like to emphasize that while we list this here for completeness we do so without the claim to have allocated funds and resources to address these immediately. Future work which we are considering focuses on more complex tasks, variable sensing ranges among the sensor nodes, and 3D environments where our algorithm is expected to self-compose or reorganize itself based on continuously changing conditions. Furthermore, the current literature on swarm-based emergency (cell-)communication networks still focuses mainly on (autonomous and decentralized) swarms consisting entirely of UAVs [25]. There is, however, other work, e.g., [28], which proposes the use of heterogeneous swarms (here UAVs and UGVs) for messaging systems. With our application (exploration of indoor environments) in mind, including stationary locations (such as e.g., shelters or fixed transmitters) or dedicated mobile hardware (such as e.g., mobile command centers, drone deployment vehicles, mobile maintenance stations, etc) into the operations of the swarm will bridge another gap to the real world and move us closer to the deployment of a real-world demonstrator of our work and the verification of our approaches. With regard to implementing such a demonstrator and deploying a swarm, there is a host of literature available suggesting approaches to address many of the practical issues posed (e.g., [103,104]). However, a number of aspects, such as e.g., legal concerns, remain. It is therefore not yet possible to report on a real-world trial but we assume that in the near future we will see more and more swarms become available and we hope that our theoretical work, presented here, will find its way into some of these. With this in mind, we suggest [17] for a survey of UAV-based communication (civil use), [16] for an overview over important issues and challenges in UAV communication networks and [105] for an overview over laws and regulations.

**Author Contributions:** Conceptualization, D.R. and A.F.I; methodology, D.R., K.E. and A.F.I.; software, D.R., K.E. and A.F.I.; validation, K.E., A.F.I., H.H. and D.R.; formal analysis, K.E., A.F.I. and H.H.; investigation, writing—review and editing, K.E., A.F.I. and H.H.; visualization, K.E., H.H. and A.F.I.; supervision, A.F.I..; project funding acquisition, D.R. and A.F.I. All authors have read and agreed to the published version of the manuscript.

**Funding:** We acknowledge the support from UAE ICT Fund grant on "Biologically Inspired Self-organizing Network Services."

**Acknowledgments:** DR and AFI acknowledge inspiring exchange with Prof. Vijay Kumar (University of Pennsylvania).

**Conflicts of Interest:** The authors declare no conflict of interest. The funders had no role in the design of the study; in the collection, analyses, or interpretation of data; in the writing of the manuscript, or in the decision to publish the results.

## Abbreviations

The following abbreviations are used in this manuscript:

| | |
|---|---|
| ACO | Ant Colony Optimization |
| CDT | Cumulative Distance Traveled |
| GA | Genetic Algorithm |
| GL | Goods Delivery/Logistics |
| INT | Surveillance |
| PAC | Percentage Area Coverage |
| PSO | Particle Swarm Optimization |
| RS | Remote Sensing |
| RTM | Real-Time Monitoring |
| SAR | Search and Rescue |
| SI | Structural Inspection |
| UAV | Unmanned Aerial Vehicle |
| WAN | Wireless Access Networks |
| WSN | Wireless Sensing Networks |

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
