# Peer review of "A Hybrid Voronoi Tessellation/Genetic Algorithm Approach for the Deployment of Drone-Based Nodes of a Self-Organizing Wireless Sensor Network (WSN) in Unknown and GPS Denied Environments"

_drones, doi:10.3390/drones4030033_

Round 1
Reviewer 1 Report
The paper is overall well written and easy to follow. The provided description of the proposed analysis is concrete and correct. As far as its content, merits and research findings are considered, it is suitable to be published in the Drones journal. However, there is a minor comment that should be addressed.
The authors should include a table summarizing all the symbols used in the manuscript in order to help the reader.
Author Response
For authors' response to Reviewer 1 please see the attached file.

Reviewer 2 Report
Authors combined GA and BISON and performed experiments on simulated data. The proposed WSN approach sets to evolve the locations of sensor nodes in unfamiliar and obstructed noisy environments.
The literature survey is comprehensive. Preliminaries are well presented. I have few minor suggestions.
- Fig 1 is identical to Fig 3 in [19]. You are expected to explain Fig 1 briefly in the text. What is RAS in Fig 1?
- Be consistent with acronym GA - lines 12-13: Genetic Algorithm (GA), line 64: Genetic Algorithms (GA). Both GA and GAs are used in the text.
- Line 193- It is interesting to see (quantitatively) how well PSO and BISON works.
- Line 389 - "As already mentioned above, it also turns out that – maybe surprisingly – adding noise improves performance,..." - There are few similar mentions in few places. You used simulated data and noise for the experiment. Explain how well your data and noise represent real drone data and real drone noise. Then, how much confident are you in your above claim when it comes to real noise?
- Line 452 - fasted - typo?
- This manuscript is too lengthy and difficult to follow. If possible try to combine relevant figures AND/OR reduce figure size (e.g. Fig 16-19) to reduce the number of pages. You can rephrase possible paragraphs (make them compact) and format the paper in compact scientific writing style.
Author Response
For authors' response to Reviewer 2 please see the attachment.

Reviewer 3 Report
The paper "A hybrid Voronoi Tessellation / Genetic Algorithm approach for the deployment of drone-based nodes of a self-organizing wireless sensor networks (WSN) in unknown and GPS denied environments " presents a method of improving the quality of a WSN deployment in an unknown environment. The abstract clarify from the beginning the goal of the work and the selected approach. The methods developed here represents extensions of the Bio-Inspired Self-Organizing Network (BISON) algorithm proposed by the authors in some previous work.
The paper is well written and clearly structured. A solid discussion of state of the art approaches is used in the first two sections to motivate the work, and to justify the selection of GA solution from several other possibilities including Ant Colony Optimization and Particle Swarm Optimization. Slightly unusual for a journal paper, the reference section list not less than 98! references. The second section presents also an overview of the original BISON algorithm with positive impact of understanding the methods development. Then two methods are proposed as GA-BISON - Fixed Nodes and GA-BISON - Conditional. They differ in manner the nodes affected by GA are selected and the time the method is applied. A detailed description of the conducted simulations is provided in the section four. The results are obtained on several evaluation metrics as the Number of Simulation Steps, the Percentage Area Coverage, and the Cumulative Distance Traveled. A standalone section propose a detailed discussion on noise coherence analysis. The paper ends with some conclusion based on the presented results and with presentation of some future work to be done. However, there are some points that needs to be improved:
- Although the number of state-of-the-art algorithms discussed is quite large, no comparison is made with any of them at the end.
- Regarding Table3, it is not clear what combination of parameters were used. The table presets only 9 combinations from the total number of possible combination between Population size(3) x Number of iteration (3) x Mutation rate (3), which is 27. (e.g. for results on Population size of {10,15,20} what values were used for the Number of iterations and Mutation rate?)
- It is not clear from the paper how the authors measure the Percentage Area Coverage in case of the environment filled with obstacles. In such environment, the sensing area for a node placed near an obstacle will be affected drastically by the obstacle.
- The authors do not discuss how the drones detects the obstacles in an unknown environment. In a practical situation, the performance of such detection (e.g. detection proximity radius, detection time, localization accuracy etc.) can also affect the method result.
Author Response
For authors' response to Reviewer 3 comments please see the attachment.

Round 2
Reviewer 3 Report
The manuscript "A hybrid Voronoi Tessellation / Genetic Algorithm approach for the deployment of drone-based nodes of a self-organizing wireless sensor networks (WSN) in unknown and GPS denied environments" was improved in accordance with most of the comments. Some of the weaknesses persist (e.g. lack of comparisons with state-of-the-art methods’ results), but in my opinion this version of the paper should be considered for publication.